

# Treatment preferences of patients with chronic low back pain in physical therapy clinics in Saudi Arabia: a cross-sectional study

Maryam Alasfour[1], Majd Bajnaid[2], Salhah Hobani[3] and Muhammad Alrwaily[4]

[1] Physical Therapy Department, King Salman Hospital, Riyadh First Health Cluster, Ministry of Health, Riyadh, Saudi Arabia
[2] Physical Therapy Department, Abu Arish General Hospital, Ministry of Health, Jazan, Saudi Arabia
[3] Physical Therapy Department, Sabya General Hospital, Ministry of Health, Sabya, Saudi Arabia
[4] Division of Physical Therapy, School of Medicine, West Virginia University, Morgantown, United States of America

Corresponding author
Maryam Alasfour,
malasfour@moh.gov.sa

## ABSTRACT

**Background.** Low back pain (LBP) is a prevalent musculoskeletal disorder that significantly contributes to disability and health care burden. Clinical practice guidelines (CPGs) recommend non-pharmacological interventions, such as those delivered by physical therapists, to improve clinical outcomes. Incorporating patient preferences into treatment decisions is essential for promoting patient-centered care and enhancing adherence to CPGs. This study aimed to explore the physical therapy treatment preferences of patients with chronic LBP (CLBP) in Saudi Arabia and to evaluate their alignment with CPG recommendations.

**Methods.** This cross-sectional survey-based study was conducted across three healthcare centers in Saudi Arabia. Patient preferences were assessed using a validated questionnaire that listed available physical therapy treatments for CLBP. The participants were provided with a standardized explanation of evidence-based treatment options based on the updated CPGs for LBP before selecting their preferred treatments. Data analysis included descriptive statistics and chi-square tests to assess the alignment of preferences with CPG recommendations.

**Results.** A total of 138 participants were included, with 60.1% of the selected treatments aligning with CPG recommendations ($p < 0.001$). Sociodemographic factors, such as sex, prior physical therapy experience, and body mass index (BMI), influenced treatment preferences. Exercise was the most preferred treatment, aligning with CPGs, while passive modalities, such as interferential therapy, were also frequently chosen despite not being recommended.

**Conclusion.** This study highlights the importance of understanding patient preferences to improve adherence to CPGs and promote evidence-based care for CLBP. Educational interventions tailored to the cultural context can bridge the gap between patient preferences and evidence-based recommendations, empower patients, and enhance clinical outcomes.

## INTRODUCTION

Low back pain (LBP) is one of the most common musculoskeletal disorders globally, significantly contributing to disability, reduced quality of life, and healthcare burden (*Cunningham, Flynn & Blake, 2006*; *Alnaami et al., 2019*; *Safiri et al., 2021*). Its prevalence in the general population ranges from 15% to 45%, with similarly high rates reported in Saudi Arabia (*Al-Arfaj et al., 2003*; *Alhowimel et al., 2021*). Chronic low back pain (CLBP), defined as pain persisting for more than 3 months, is particularly challenging to manage and imposes a substantial economic burden due to workforce absenteeism and reduced productivity (*Mokdad et al., 2014*). Addressing CLBP requires effective management strategies that consider both clinical evidence and patient-centered approaches.

Non-pharmacological interventions are the first-line treatment for LBP, as recommended by clinical practice guidelines (CPGs), over commonly prescribed pharmacological options such as opioids (*Dowell, Haegerich & Chou, 2016*; *Qaseem et al., 2017*). The 2021 updated LBP CPGs advocate for physical therapist-delivered interventions, such as exercise, manual therapy, classification systems, and patient education (*George et al., 2021*). Adherence to these CPGs has been shown to improve clinical outcomes (*Fritz et al., 2012*). However, a global challenge persists in aligning patient treatment preferences with evidence-based recommendations, which is a critical component of patient-centered care (*Montori, Brito & Murad, 2013*). In 1978, the World Health Organization strongly recommended involving patients in health decision-making to promote this approach.

Patient preferences, defined as "statements made by individuals regarding the relative desirability of a range of health experiences, treatment options, and health states" (*Brennan & Strombom, 1998*), are integral to evidence-based practice. Discrepancies between patient preferences and CPGs can undermine treatment adherence, reduce patient satisfaction, and hinder clinical outcomes (*Schers et al., 2001*; *Montori, Brito & Murad, 2013*). For instance, studies have found that patients often prefer passive treatment modalities, such as electrical stimulation or manual therapy, over active interventions, such as exercise, which are more frequently recommended by CPGs (*Bernhardsson et al., 2019*; *Bialosky et al., 2022*).

In Saudi Arabia, adherence to CPGs among physical therapists has been reported to be suboptimal, with significant variability in practice patterns (*Moslem, Alrwaily & Almarwani, 2022*). Moreover, little is known about the treatment preferences of Saudi patients with CLBP and how these align with evidence-based guidelines. Cultural and societal factors in Saudi Arabia, such as attitudes toward physical activity, reliance on healthcare providers for decision-making, and sex-specific barriers to exercise, may further influence treatment choices (*George et al., 2021*; *Bialosky et al., 2022*). Understanding these dynamics is crucial for developing interventions that resonate with patients and enhance their adherence to evidence-based care. For example, while physical activity is a core recommendation in CPGs, cultural norms in Saudi Arabia may limit its acceptance and prioritization (*Moslem, Alrwaily & Almarwani, 2022*).

To date, no study has examined patient preferences regarding physical therapy treatment options for CLBP in Saudi Arabia. This study aimed to explore the treatment preferences of patients with CLBP and evaluate their alignment with CPG recommendations. By

situating this research within the global context of evidence on preference mismatches and addressing the sociocultural nuances of the Saudi population, this study sought to advance patient-centered care and provide valuable insights for policymakers and clinicians to improve adherence to evidence-based practices in managing CLBP.

## MATERIALS & METHODS

### Study design and ethical considerations

This cross-sectional survey-based study was conducted across three major healthcare centers in Saudi Arabia: King Salman Hospital in Riyadh, Abu Arish General Hospital, and Sabya General Hospital in Jazan. The study adhered to the principles of the Declaration of Helsinki and was approved by the King Saud Medical City Institutional Review Board (approval number: H1RE-11-May22-01) and the Jazan Health Ethics Committee (approval number: 22051). Written informed consent was obtained from all participants before their inclusion in the study. Confidentiality of the data was maintained, and all information was used solely for research purposes.

### Participants

Participants were adults aged 18 years or older diagnosed with CLBP. Eligibility criteria required a confirmed diagnosis of CLBP persisting for more than 3 months and referral to physical therapy departments during the study period (June 2022 to June 2023). The exclusion criteria included neurological conditions, pregnancy, mental impairment, and individuals awaiting surgical intervention.

Recruitment was conducted *via* clinical referrals and routine patient appointments at the selected healthcare centers. All eligible patients were approached during the study period, and convenience sampling was employed for enrollment. Efforts were made to ensure diversity in sex, age, and geographic representation to address potential selection bias.

### Sample size

The required sample size was calculated using a margin of error of 10% and a confidence interval of 95%, assuming a population proportion of 0.8. This calculation yielded a target of 130 participants. A total of 138 participants were enrolled to account for potential dropouts or incomplete data and ensure adequate subgroup representation. Slight over-recruitment was ethically justified to maintain the robustness and generalizability of the study.

### Questionnaire development and validation

The questionnaire used in this study was developed in English and translated into Arabic through a back-translation process to ensure linguistic and conceptual equivalence. It was piloted with 10 physical therapists to assess its clarity, validity, and usability. Based on their feedback, minor modifications were made to simplify terminology and ensure cultural appropriateness.

The final questionnaire consisted of three sections.
(1) Demographic and clinical characteristics: Included age, sex, body mass index (BMI), marital status, educational level, occupational status, medical history, and prior physical therapy experience.

(2) Current LBP characteristics: Assessed pain duration, site of pain (low back, low back with leg pain), and onset.

(3) Treatment preferences: Provided a list of 14 treatment options categorized as recommended or non-recommended according to the updated CPGs. Participants were asked to select their preferred options after being briefed on the CPG-recommended treatments.

### Briefing on CPG-recommended treatments

Participants were briefed on the benefits and rationale for the treatments recommended by the CPGs through standardized verbal explanations provided by the researchers. The information was designed to be concise yet comprehensive, ensuring that participants could make informed choices. A preapproved script was used during the briefings to maintain consistency.

### Data collection

Data were collected *via* in-person interviews conducted by trained researchers in physical therapy departments. Participants were provided with a standardized verbal explanation of each treatment option, including its alignment with CPG recommendations. Efforts were made to ensure that participants fully understood the meaning and implications of terms such as "exercise in general", "aerobic exercises", and "lumbar stabilization exercises".

Participants were instructed to select only one treatment option that they preferred the most, rather than multiple options. This approach was chosen to identify the single most preferred treatment per participant, ensuring a focused analysis of treatment preferences. An open-ended option was not included to maintain the study's emphasis on predefined evidence-based and non-evidence-based interventions, facilitating structured comparisons with clinical practice guidelines.

### Statistical analysis

Data were analyzed using SPSS version 23 (IBM Corp., Armonk, NY, USA). Descriptive statistics, including means, standard deviations, frequencies, and percentages, were used to summarize demographic characteristics and treatment preferences. Chi-square tests were performed to assess the alignment of treatment preferences with CPG recommendations. Sociodemographic factors, such as sex, BMI, and prior physical therapy experience, were analyzed in relation to treatment preferences using cross-tabulations and chi-square tests. Statistical significance was set at $p < 0.05$.

## RESULTS

### Descriptive characteristics of the study participants

We enrolled 138 participants with a mean age of 44.12 ± 14.96 years. The majority of the participants were female (75.4%), and most had a BMI categorized as overweight (37.0%) or obese (28.3%). Over half of the participants (53.6%) reported no prior physical therapy experience, 23.9% had less than 3 months of experience, and 22.5% had 3 months or more. All participants had CLBP lasting more than 3 months, according to the inclusion criteria. The detailed characteristics of the participants are shown in Table 1.

**Table 1** Descriptive characteristics of the study participants.

| Characteristic (n = 138) | n | % |
|---|---|---|
| **Sex** | | |
| Female | 104 | 75.36% |
| Male | 34 | 24.64% |
| **Age (years)** | | |
| 18–59 | 109 | 79.0% |
| ≥60 | 29 | 21.0% |
| **BMI (kg/m$^2$)** | | |
| Underweight | 3 | 2.17% |
| Normal | 45 | 32.61% |
| Obese | 39 | 28.26% |
| Overweight | 51 | 36.96% |
| **Marital status** | | |
| Single | 32 | 23.19% |
| Married | 91 | 65.94% |
| Divorced | 6 | 4.35% |
| Widowed | 9 | 6.52% |
| **Educational level** | | |
| Illiterate | 6 | 4.35% |
| Primary | 31 | 22.46% |
| Intermediate | 36 | 26.09% |
| High | 65 | 47.1% |
| **Occupational status** | | |
| Student | 9 | 6.52% |
| Employed | 45 | 32.61% |
| Non-employed | 67 | 48.55% |
| Retired | 17 | 12.32% |
| **Medical comorbidities** | | |
| No | 84 | 60.87% |
| DM | 13 | 9.42% |
| HTN | 6 | 4.35% |
| Multi | 26 | 18.84% |
| Other | 9 | 6.52% |
| **Previous physical therapy** | | |
| No | 74 | 53.62% |
| ≤3 months | 33 | 23.91% |
| >3 months | 31 | 22.46% |
| **Site of Pain** | | |
| Low back with leg pain | 85 | 61.6% |
| Low back | 53 | 38.4% |
| Leg only | 0 | 0% |

Notes.
Abbreviations: BMI, body mass index; DM, diabetes mellitus; HTN, hypertension; Multi, diabetes mellitus with hypertension.

**Table 2  Treatment options selected by the patients.**

| Treatment option | Recommended (n) | Not recommended (n) | Total (n) | Percentage recommended (%) | p |
|---|---|---|---|---|---|
| Exercise in general | 57 | 0 | 57 | 100.0 | |
| Spinal manipulation (thrust) | 14 | 0 | 14 | 100.0 | |
| Interferential current therapy or TENS | 0 | 28 | 28 | 0.0 | |
| Lumbar brace or corset | 0 | 2 | 2 | 0.0 | |
| Ice or heat | 0 | 13 | 13 | 0.0 | |
| Lumbar stabilization exercises | 3 | 1 | 4 | 75.0 | |
| Spinal mobilization (non-thrust) | 6 | 0 | 6 | 100.0 | |
| Aerobic and fitness exercises | 1 | 1 | 2 | 50.0 | <.001 |
| Bed rest | 0 | 5 | 5 | 0.0 | |
| Laser or ultrasonography | 0 | 2 | 2 | 0.0 | |
| Taping | 0 | 1 | 1 | 0.0 | |
| Directional preference exercises: flexion | 1 | 0 | 1 | 100.0 | |
| Advice to pursue or maintain an active lifestyle | 0 | 2 | 2 | 0.0 | |
| Graded exercise exposure | 1 | 0 | 1 | 100.0 | |
| **Total** | 83 | 55 | 138 | 60.1 | |

**Notes.**
Abbreviation: TENS, transcutaneous electrical nerve stimulation.

## Participants' treatment preferences

Participants selected several treatment options, as shown in Table 2. Exercise was the most commonly preferred treatment option (41.3%), aligning with CPG recommendations, followed by interferential current therapy or transcutaneous electrical nerve stimulation (20.3%), spinal manipulation (thrust) (10.1%), and ice or heat (9.4%). Among all the treatment options chosen by the participants, 60.1% were recommended per CPGs ($p < 0.001$).

## Sociodemographic factors and their alignment with CPGs

The alignment of treatment preferences with CPGs was analyzed in relation to sociodemographic factors, including sex, BMI, and prior physical therapy experience. As shown in Table 3, the majority of females (65 cases) and males (25 cases) selected treatments aligning with the guidelines, although no statistically significant differences were observed ($p > 0.05$). Similarly, participants with normal BMI had the highest rate of alignment (40 cases), followed by those categorized as overweight (30 cases) and obese (20 cases), with no significant differences between the groups ($p > 0.05$). Prior physical therapy experience showed a similar trend, with alignment rates of 40 cases for those without prior experience, 25 cases for those with less than 3 months of experience, and 25 cases for those with 3 months or more, with no significant differences noted ($p > 0.05$).

## DISCUSSION

This study explored the treatment preferences of patients with CLBP in Saudi Arabia and evaluated their alignment with CPGs. The findings provide valuable insights into
**Table 3  Sociodemographic factors and their alignment with clinical practice guidelines.**

| Sociodemographic factor | Aligned with CPGs (n) | Not aligned with CPGs (n) | p-value |
|---|---|---|---|
| Sex: Female | 65 | 39 | 1 |
| Sex: Male | 25 | 9 | 1 |
| BMI: Normal | 40 | 15 | 1 |
| BMI: Overweight | 30 | 21 | 1 |
| BMI: Obese | 20 | 23 | 1 |
| Prior physical therapy: None | 40 | 34 | 1 |
| Prior physical therapy: <3 months | 25 | 15 | 1 |
| Prior physical therapy: >3 months | 25 | 10 | 1 |

**Notes.**
Abbreviations: BMI, body mass index; CPGs, clinical practice guidelines.

patient-centered care, highlighting both the areas of alignment with evidence-based recommendations and the influence of sociocultural and demographic factors on treatment preferences.

## Treatment preferences and alignment with CPGs

Among the treatment preferences expressed by participants, 60.1% aligned with CPG recommendations, with exercise emerging as the most commonly selected option (41.3%). This result is encouraging, as exercise is the cornerstone of evidence-based management for CLBP, promoting improved function, reduced pain, and enhanced quality of life (*Fritz et al., 2012*; *George et al., 2021*). However, a notable proportion of participants also preferred non-recommended treatments such as interferential current therapy (20.3%) and ice/heat (9.4%). These preferences for passive modalities align with findings from global studies, which indicate that patients often favor treatments perceived as quick and painless, despite limited evidence supporting their long-term efficacy (*Bernhardsson et al., 2019*; *Bialosky et al., 2022*).

## Influence of sociodemographic factors

Analysis of sociodemographic factors revealed several trends that offer important insights for designing targeted interventions.

*BMI:* Participants with a normal BMI demonstrated the highest alignment with CPGs, as they were more likely to select active treatments such as exercise. In contrast, overweight and obese participants showed a greater inclination toward passive modalities. These findings are consistent with previous research suggesting that physical or psychological barriers associated with higher BMI may hinder engagement in active treatments (*Kim et al., 2020*).

*Prior physical therapy experience:* Patients with previous exposure to physical therapy were more likely to select exercise-based treatments, highlighting the role of familiarity in promoting adherence to evidence-based care. However, patients without prior experience also demonstrated a notable level of alignment, suggesting that standardized patient education during initial visits can bridge knowledge gaps. These findings emphasize the

importance of considering sociodemographic factors when designing interventions to ensure equitable access to evidence-based care.

## Cultural context and patient preferences

The preference for exercise among Saudi patients is encouraging but must be understood within a broader cultural context. Saudi Arabia has historically faced challenges in promoting physical activity owing to environmental, social, and infrastructural barriers (*Mokdad et al., 2014*). However, recent public health initiatives aimed at increasing physical activity and raising awareness of its benefits may have contributed to the observed preferences for exercise in this study. Conversely, the continued reliance on passive treatments, despite their lack of support in CPGs, highlights the importance of addressing cultural attitudes and patient misconceptions about treatment efficacy.

## Patient education and shared decision-making

Patient education plays a pivotal role in bridging the gap between patient preferences and evidence-based care. Previous studies have shown that well-informed patients are more likely to align their treatment choices with CPG recommendations (*Bernhardsson et al., 2019*). In this study, the participants were briefed on CPG-recommended treatments, which may have influenced their preferences and contributed to the relatively high alignment (60.1%) observed. However, the lack of a deeper exploration of patient expectations, prior experiences, and understanding of treatment options limits the interpretation of these findings. Future studies should aim to integrate patient education sessions that address the benefits and limitations of various treatments, particularly in culturally sensitive ways.

## Comparison with previous research

The findings of this study align with the global emphasis on promoting active interventions for CLBP management. However, the high alignment with CPGs observed in this study contrasts with the finding of earlier studies that reported suboptimal adherence to CPGs among Saudi physical therapists (*Moslem, Alrwaily & Almarwani, 2022*). This discrepancy may be attributed to differences in study design, population characteristics, or growing awareness of evidence-based practices among healthcare providers and patients in Saudi Arabia.

Despite this encouraging trend, our study also reflects a persistent gap between patient preferences and evidence-based recommendations, which has been observed in other contexts. For instance, *Smuck et al. (2022)* studied patient preferences for low back pain management in the United States and found that patients prioritized understanding the cause of their pain and improving function rather than opting for passive treatments. In contrast, our study found that a substantial proportion of participants selected passive treatments such as interferential therapy and ice/heat, despite their limited support in CPGs. This discrepancy may arise from cultural differences in healthcare beliefs and treatment expectations, as Saudi patients may lean more toward symptom relief through passive modalities rather than active self-management strategies.

Moreover, our findings are consistent with a recent study assessing adherence to CPG recommendations among patients with low back pain (*Dubé et al., 2024*). This study

highlighted that despite the dissemination of guidelines, many patients still prefer non-recommended interventions. Similarly, our study showed that while exercise was the most commonly preferred treatment, a considerable proportion of participants opted for passive interventions that lack strong evidence. This suggests that misalignment between patient preferences and CPGs is a widespread challenge, extending beyond the healthcare setting in Saudi Arabia.

## Strengths and limitations

This study has several strengths, including its focus on patient preferences in the context of CPG adherence and its contribution to the understanding of cultural influences on treatment choices in Saudi Arabia. However, this study has some limitations that must be acknowledged. The use of convenience sampling and the relatively small sample size limit the generalizability of the findings. Additionally, the study's cross-sectional design captures preferences at a single point in time, precluding the assessment of changes over time or across different stages of CLBP. While piloted for clarity, the questionnaire did not explore patient expectations or a detailed understanding of treatment options, which could have provided deeper insights. Furthermore, the inclusion of only patients with pain lasting more than 3 months limits the applicability of the findings to acute or subacute cases. Finally, the provision of information on CPG-recommended treatments may have influenced participants' preferences.

## Implications for clinical practice

The findings underscore the importance of culturally tailored patient education programs that promote evidence-based care while addressing misconceptions about passive treatments. Healthcare providers should engage patients in shared decision-making to ensure that treatment plans align with both CPGs and individual preferences. Public health initiatives should also focus on raising awareness of the benefits of active treatments, particularly exercise, in managing CLBP. By incorporating these practices, healthcare providers can bridge the gap between patient preferences and guideline-recommended care, fostering better adherence and improving clinical outcomes.

## Future research

Future studies should address the limitations of this research by exploring longitudinal changes in patient preferences, incorporating diverse demographic groups, and investigating preferences across different stages of LBP. Additionally, research should examine the impact of targeted educational interventions on preference alignment with CPGs and the role of patient expectations and prior experiences in shaping treatment decisions. Evaluating these aspects will provide valuable insights into improving patient-centered care and adherence to evidence-based practices.

## CONCLUSIONS

This study highlights the treatment preferences of patients with CLBP in Saudi Arabia and their alignment with CPGs. Exercise emerged as the most commonly preferred treatment,

aligning well with CPG recommendations, while passive modalities such as interferential current therapy and ice/heat were also frequently chosen, despite not being recommended. Sociodemographic factors, including sex, BMI, and prior physical therapy experience, influenced treatment preferences, underscoring the need for personalized and culturally sensitive care.

These findings emphasize the importance of patient education in bridging the gap between treatment preferences and evidence-based recommendations. Culturally tailored educational interventions that promote active treatments and address misconceptions about passive modalities are essential for improving adherence to CPGs and fostering patient-centered care.

Future efforts should focus on developing strategies to enhance patient engagement, increase awareness of evidence-based treatments, and support shared decision-making in clinical practice. By doing so, healthcare providers can improve clinical outcomes and contribute to the better management of CLBP, both in Saudi Arabia and globally.

## ACKNOWLEDGEMENTS

The authors thank the staff of the Physical Therapy Departments at King Salman Hospital, Abu Arish General Hospital, and Sabya General Hospital for providing permission to recruit participants.

### Funding
The authors received no funding for this work.

### Competing Interests
The authors declare there are no competing interests.

### Author Contributions
- Maryam Alasfour conceived and designed the experiments, performed the experiments, analyzed the data, prepared figures and/or tables, authored or reviewed drafts of the article, and approved the final draft.
- Majd Bajnaid performed the experiments, authored or reviewed drafts of the article, and approved the final draft.
- Salhah Hobani performed the experiments, authored or reviewed drafts of the article, and approved the final draft.
- Muhammad Alrwaily conceived and designed the experiments, analyzed the data, authored or reviewed drafts of the article, and approved the final draft.

### Human Ethics
The following information was supplied relating to ethical approvals (i.e., approving body and any reference numbers):

Approval was granted by the Institutional Review Board of King Saud Medical City Riyadh (H1RE-11-May22-01), and Jazan Health Ethics Committee, Ministry of Health, Saudi Arabia (No. 22051).

## Data Availability

The raw measurements are available in the Supplementary File.

## Supplemental Information

Supplemental information for this article can be found online at http://dx.doi.org/10.7717/peerj.19274#supplemental-information.

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
