# Peer review of "Treatment preferences of patients with chronic low back pain in physical therapy clinics in Saudi Arabia: a cross-sectional study"

_PeerJ, doi:10.7717/peerj.19274_

## Round 0.1 · original submission · Major Revisions

Please address the detailed comments of the reviewers.

·

Basic reporting

The Research title reflects the scope of the study.
Generally, the manuscript was written in good English. However, the authors are required to check numerous redundancies in the abstract and the in main text.
Some of the phrasing in the Introduction can be further improved to emphasized the gaps for conducting the study. The authors have provided some citations (literature review) but lacks depth in justifying the study's importance.
The authors should expand on the mismatch between patient preferences and adherence to clinical guidelines, supported by global evidence.
Also to provide a clearer rationale for the study in Saudi Arabia, highlighting gaps in existing literature.
With regards to overall structures and presentation of tables: the authors are advised to revise the presentation of the methodology and results. The Tables require major revision, as to present them according to common standards and not directly copying from the SPSS output.
Table 1 is detailed but underutilized in the analysis. Discuss how sociodemographic factors (e.g., gender, BMI, or prior physical therapy experience) influenced preferences.
Table 2 clearly categorizes preferences but lacks context. For example, why were certain treatments preferred over others? Include a brief summary of notable findings.
Figures/bar charts may be used to visualize the treatment preferences showing the percentage alignment with CPGs to enhance readability.
The Introduction needs to better articulate the hypothesis and knowledge gap.
The Discussion repeats points made in the Results and lacks an interpretation of why certain preferences were observed. For example, why is exercise preferred over interferential therapy? Tie this to cultural or clinical contexts.
The raw data is mentioned but not sufficiently discussed in the manuscript. The authors should:
While the authors report that 60.1% of preferences align with CPGs, they do not hypothesize or interpret why this is significant.
The discussion must be reframed to address how the findings contribute to patient-centered care or adherence to evidence-based guidelines.
Need to revise limitations and conclusions as they seem redundant.
Include limitations related to the sample size and convenience sampling method, as these affect generalizability.

Experimental design

1. While the research question is clear, but the study does not explicitly articulate how filling this gap advances patient-centered care or informs policy changes in Saudi Arabia.
2. Lacks details on how participants were referred or recruited.
3. Were patients actively approached by physical therapists, or was this based on routine clinical referrals?
4. Clarify whether all eligible patients during the study period were approached or if selection criteria were applied.
5. The methodology states that the required sample size was 130, yet 138 participants were enrolled. This discrepancy raises ethical concerns about why more participants were included than calculated. Address why this over-recruitment occurred and justify it ethically (e.g., anticipated dropouts, ensuring adequate subgroup representation).
6. The study claims convenience sampling, which has inherent biases: Acknowledge and discuss how this might affect the generalizability of the findings.
7. The methodology lacks details on the validity and reliability of the questionnaire used. Was it previously validated, or was it developed for this study? If the latter, describe the validation process (e.g., pilot testing results or expert review).
8. Include an explanation of how the questionnaire measured preferences (e.g., Likert scale, multiple-choice format) and the rationale behind its design.
9. Specify how and where data was collected (e.g., in-person interviews, online surveys) and by whom (e.g., researchers or clinic staff).
10. Describe how the participants were briefed about CPG-recommended treatments. Was it a verbal explanation, or was written material provided? Who delivered this information, and how was standardization ensured?
11. While demographic data collection is mentioned, the methodology does not explain how these variables were chosen or their potential role in influencing preferences.
12. Statistical tests are briefly mentioned (simple frequency analysis and P-values), but the rationale for using these tests and the software parameters (e.g., SPSS version 23) is not elaborated; Include justification for the statistical methods, particularly for testing the alignment of preferences with CPGs (e.g., chi-square tests).

Validity of the findings

1. The results were presented but lacked details description.
2. Several sociodemographic characteristics were presented but were not justified, to why they are relevant in the study.
3. The presentation of tables needs improvement, not to directly copy from statistical test output.
4. The discussion lacks comparisons with other studies and an explanation of why the findings are as such. For instance, why exercises are the most preferred among the participants, and so on.
5. The limitations did not directly suggest the weaknesses or strengths of the study.
6. The conclusion did not conclude the findings based on the research objectives, instead, it is a repetition of the limitations.

Additional comments

The manuscript requires substantial amendment to show the novelty of the study, with major improvements should be focused on the methodology, results and discussion.

·

Basic reporting

The article is written with professional standards of courtesy and expression in clear, unambiguous, and technically correct English.
The literature references are sufficiently provided.
The article is written professionally.

Experimental design

The original primary research lies within the aims and scope of the journal.
The research question is well-defined, relevant, and meaningful. The research gap has been justified.
The investigation aligns with the research question.
The methodology is described in sufficient detail.

Validity of the findings

No impact or novelty is assessed.
Yes, all underlying data have been provided; they are robust, statistically sound, and controlled.
Conclusions are well stated, linked to the original research question, and limited to supporting results

Additional comments

The article is written with sufficient details.

Reviewer 3 ·

Basic reporting

Despite the persistent nature of the problem (CLBP), the study only focused on treatment choices at the time of the survey, making no effort to investigate patients' perspectives on their expectations and preferences towards recovery.

Experimental design

• Before directly asserting that Saudi Arabia has not yet examined patient preferences for physical therapy treatment options for CLBP, please provide data from any other studies that have examined this objective elsewhere, along with the results. Provide the economic and workforce losses resulting from CLBP in the KSA, if available, and explain why you believe a country-specific study is necessary for this objective. Firstly, before proceeding with this objective, please add a segment on Saudi PTs adherence to CPGs.
• The section on questionnaire development is very short, and the provided information wouldn’t permit replication of this work. I suggest the authors broaden this section and elucidate the methods used to achieve validity. What outcomes did the piloting result in? Were the questions developed in Arabic or English?
• The procedure section is not clear. Were the participants permitted to select all the options, or was only one option per patient permitted? Why did the study participants not receive an open-ended option?
• Did the patients understand the meaning of the different exercises listed on the supplementary table such as general exercises, aerobic exercises, lumbar stabilization exercises, and others?
• Why did the authors fail to enquire about the patients' expectations and views on diagnosis, including radiological investigations? The authors did not attempt to identify any relationships between the factors under study.
• It's challenging to determine which subcategory of patients opted to exercise based on the tables.
• I was unable to determine the duration of pain in the table. When the adherence to CPGs is so high, why do the authors think they develop chronicity of symptoms? Do they think the preference would be different in acute LBP patients?
• Does Saudi culture readily encourage exercise as an intervention? Please provide insight into the social and cultural factors that influence treatment choices. Further, what’s the author's take on why Saudi patients prefer exercises?
• The overall study provides an impression that the treatment recommendations are moving in the right direction in KSA, contradicting the Moslem WM, Alrwaily M, Almarwani MM. Adherence to low back pain clinical practice guidelines by Saudi physical therapists: a cross-sectional study. Physiother Theory Pract. 2022;38(7):938-951. doi:10.1080/09593985.2020.1806420 This needs addressing.
• The discussion fails to consider the perspectives of other studies where patients have denounced exercises.
• The manuscript does not mention the strengths and weaknesses of the study.

Validity of the findings

The authors' recommendations for appropriate care during the survey may have primarily influenced patient preferences. This may have impacted the study outcomes. It is difficult to discern the reasoning behind the patients' choices. Given the unpredictable prognosis associated with CLBP, I speculate if prior experiences had any influence on their decisions. Provided the lack of widely established therapeutic approaches for managing CLBP, this appears to be an overly reductionist concept.

Additional comments

None

---

## Round 0.2 · Minor Revisions

Your submission still needs some more minor revisions

Reviewer 3 ·

Basic reporting

The authors have addressed all the major concerns raised during the initial review process. The manuscript could, however, be further strengthened if the authors could compare and contrast the results of the current study with other works looking into similar objectives.

Experimental design

The methodological design is satisfactory, and the manuscript effectively addresses the procedures.

Validity of the findings

Considering the limitations of cross-sectional design and the fact that the researchers were briefed on the benefits and rationale for the treatments recommended by the CPGs prior to the study, I believe this could affect the validity of the results.

---

## Round 0.3 · accepted · Accept

Dear Authors,

Congratulations!

Reviewer 3 ·

Basic reporting

The authors have effectively addressed all issues, and the material appears satisfactory and may be approved for publication.

Experimental design

Satisfactory. No further comments.

Validity of the findings

Limitations indicated. No further comments.